# Epigenetics is all you need: A transformer to decode chromatin structural compartments from the epigenome

Esteban Dodero-Rojas[1][�], Angel Mendieta[1,2][�], Yao Fehlis[3], Nicholas Mayala[3], Vinícius G. Contessoto[1,4]*, José N. Onuchic[1,4,5,6]*

**1** Center for Theoretical Biological Physics, Rice University, Houston, Texas, United States of America, **2** School of Physics, Universidad de Costa Rica, San Pedro de Montes de Oca, San José, Costa Rica, **3** Research and Advanced Development, Advanced Micro Devices, Austin, Texas, United States of America, **4** Department of Physics and Astronomy, Rice University, Houston, Texas, United States of America, **5** Department of Chemistry, Rice University, Houston, Texas, United States of America, **6** Department of Biosciences, Rice University, Houston, Texas, United States of America

l These authors contributed equally to this work.
* contessoto@rice.edu (VGC); jonuchic@rice.edu (JNO)

## Abstract

Chromatin within the nucleus adopts complex three-dimensional structures that are crucial for gene regulation and cellular function. Recent studies have revealed the presence of distinct chromatin subcompartments beyond the traditional A/B compartments (eu- and hetero-chromatin), each exhibiting unique structural and functional properties. Here, we introduce TECSAS (Transformer of Epigenetics to Chromatin Structural AnnotationS), a deep learning model based on the Transformer architecture, designed to predict chromatin subcompartment annotations directly from epigenomic data. TECSAS leverages information from histone modifications, transcription factor binding profiles, and RNA-Seq data to decode the relationship between the biochemical composition of chromatin and its 3D structural behavior. TECSAS achieves high accuracy in predicting subcompartment annotations and reveals the influence of long-range epigenomic context on chromatin organization. Furthermore, we demonstrate the model's capability to predict the association of loci with nuclear bodies, such as the lamina, nucleoli, and speckles, providing insights into the role of these structures in shaping the 3D genome organization. This study highlights the potential of deep learning models for deciphering the complex interplay between epigenomic features and 3D genome organization, allowing us to better understand genome structure and function.

## Author summary

The three-dimensional structure of DNA in the cell nucleus influences gene regulation and expression. Changes in this structure can contribute to how genes

**Data availability statement:** All relevant data are within the manuscript and its Supporting information files.

**Funding:** This research was supported by the Center for Theoretical Biological Physics, sponsored by the NSF (Grants PHY-2019745 and PHY-2210291 to JNO) and by the Welch Foundation (Grant C-1792). JNO is a Cancer Prevention and Research Institute of Texas (CPRIT) Scholar in Cancer Research. The funders had no role in study design, data collection and analysis, decision to publish, or preparation of the manuscript.

**Competing interests:** The authors have declared that no competing interests exist.

are activated or silenced, which is essential for how cells work. We developed a computational tool called TECSAS to predict compartment annotations of chromosomes from epigenomic data. TECSAS does not rely on contact maps but uses only epigenetic marks to define chromatin states. Our results show that TECSAS also predicts chromosome regions associated with nuclear structures like interactions with lamina, nucleoli, and speckles. This indicates that epigenetic data may carry sufficient information to capture the genome's 3D organization heterogeneity among different cell types.

## 1 Introduction

Within the eukaryotic cell nucleus, the genome folds into three-dimensional structures that vary depending on cell type and stage of development [1]. These architectural features play a crucial role in regulating gene expression, and disruptions in this organization have been linked to various diseases [2–5]. Over the past decade, DNA-DNA proximity ligation assays, such as Hi-C [6–10], have enabled the systematic study of genome organization by measuring the frequency of chromatin contacts throughout the genome. Hi-C experiments have revealed that chromatin segregates into regions with preferential long-range interactions, known as compartments [10]. A-type compartments are gene-rich and associated with active and less dense chromatin (euchromatin). These compartments are enriched with proteins like RNA polymerase and specific histone modifications, such as H3K4me3. In contrast, B-type compartments are gene-poor and linked to inactive and more dense chromatin (heterochromatin). They are often associated with the enrichment of different histone modifications, such as H3K9me3 and H3K27me3 [11].

High-resolution Hi-C experiments have revealed that chromatin exhibits finer compartmentalization than the A and B [11]. For instance, within B compartments, specific regions are prone to interact with the nuclear lamina or nucleoli. These observations led to the concept of subcompartments, which further classify chromatin based on distinct structural and functional properties. Rao et al. (2014) [11] demonstrated that five subcompartments (A1, A2, B1, B2, and B3) effectively capture the structural heterogeneity observed in Hi-C experiments on the human lymphoblastoid cell line GM12878. Each subcompartment exhibits a unique enrichment profile of epigenetic marks, such as histone modifications. For example, B2 and B3 subcompartments show depletion of most histone modifications, while B1 shows neither depletion nor enrichment of histone modifications except for H3K27me3. Additionally, subcompartment identity correlates with the binding of specific nuclear lamina and nucleoli-associated proteins, suggesting a link between structural diversity and interactions with nuclear bodies [8,12].

The identification of chromatin compartments and subcompartments has initially relied on the analysis of Hi-C data, which provides information about the spatial proximity of genomic regions and laid the foundation for multiple chromatin theoretical models [13–17]. Several computational methods have been developed to classify regions of the genome into these structural categories based on patterns observed

in Hi-C contact maps [18–20]. For example, the SNIPER method focuses on predicting subcompartments from moderate-coverage Hi-C data by imputing inter-chromosomal contacts [18]. The algorithm Calder uses the Hi-C intrachromosomal interactions to identify multi-scale chromatin subcompartments and compartment domains that enable analysis at variable data resolutions [19]. Recent efforts have focused on linking epigenetic information, such as histone modifications and transcription factor binding, to chromatin compartments and subcompartments labeling. Similar to Calder, a deep learning method, called SLICE, generates subcompartment annotations at 25, 50 and 100 kilobase resolution from Hi-C maps. Interestingly, SLICE provides structural annotations ranging from 2 to 12 possible states [21]. Additionally, the reliance on Hi-C data for training or validation in these methods restricts their applicability to cell types with available Hi-C experiments. Therefore, developing methods to predict chromatin organization directly from epigenomic data is crucial for expanding our understanding of 3D genome structure across diverse cell types.

Though trained partially on Hi-C map information, CoRNN [20], a deep learning model based on recurrent neural networks, utilizes histone modification data to predict A/B compartments in different cell lines. Epiphany tool also employs a deep learning model to predict cell-type-specific Hi-C contact maps from 1D epigenomic signals, which could be used to label compartments and subcompartments for each locus [22]. Additionally, based only on the epigenome data and not using Hi-C maps, PyMEGABASE (PYMB) uses ChIP-Seq from Histone Modification and Transcription factor, and RNA-Seq to predict compartments and subcompartments for hundreds of cell types [23]. PYMB's interpretable predictions and transferability across cell types and species further demonstrate the potential of data-driven models for understanding 3D genome organization. Notwithstanding, PYMB is based on the MEGABASE framework [24]. MEGABASE uses a physics-based approach similar to the Potts Model that builds an energy function focused on the association between epigenetic marks and subcompartments [16,17,24,25]. The potential for using the complex interplay between multiple epigenetic marks to identify structural annotations has not been explored.

Moreover, recent work has shown great success in predicting phenotypic features from DNA-sequence (CpGPT [26]) and methylation profiles (MethylGPT [27]) - highlighting the promise of deep-learning architectures, such as Evo models [28,29], to predict highly complex trend from genomic data. This study introduces a novel approach to predicting chromatin's structural annotations based on the epigenome (e.g., histone modifications, transcription factor binding, RNA expression). We introduce TECSAS (Transformer of Epigenetics to Chromatin Structural AnnotationS), a deep learning model that leverages the power of Transformers and Attention layers to capture complex relationships between various epigenetic marks and predict subcompartment annotations with high accuracy [30]. Unlike other methods that rely on Hi-C data for training or validation, our approach focuses solely on epigenetic information. This allows us to predict subcompartment annotations even in cell types where Hi-C data is unavailable. Additionally, our results demonstrate that TECSAS versatility allows for the prediction of additional structural features, such as the association of loci with nuclear bodies like the lamina, nucleoli, and speckles, by simply fine-tuning the final layer of the model. TECSAS flexibility enables the exploration of diverse aspects of 3D genome organization using a single, unified framework.

## 2 Results

### 2.1 TECSAS predicts subcompartments by decoding the loci context of the epigenetic profile

This study introduces TECSAS (Transformer of Epigenetics to Chromatin Structural Annotations), a deep learning model based on the Transformer architecture, to predict structural information from 1D epigenomic data. Fig 1 summarizes the workflow of TECSAS. The model takes as input the signal intensity of various epigenomic features locus-wise, including RNA-seq, histone modification ChIP-seq, and transcription factor ChIP-seq experiments. This diverse data set represents the DNA's biochemical composition and transcriptional activity. To provide context and capture long-range dependencies along the genome sequence, each locus is characterized by the signal intensity of these epigenomic features within a defined neighborhood of N loci upstream and downstream. We refer to this combined input as the "epigenomic profile" of the locus. TECSAS aims to learn the relationship between a locus' structural annotation and its corresponding epigenomic

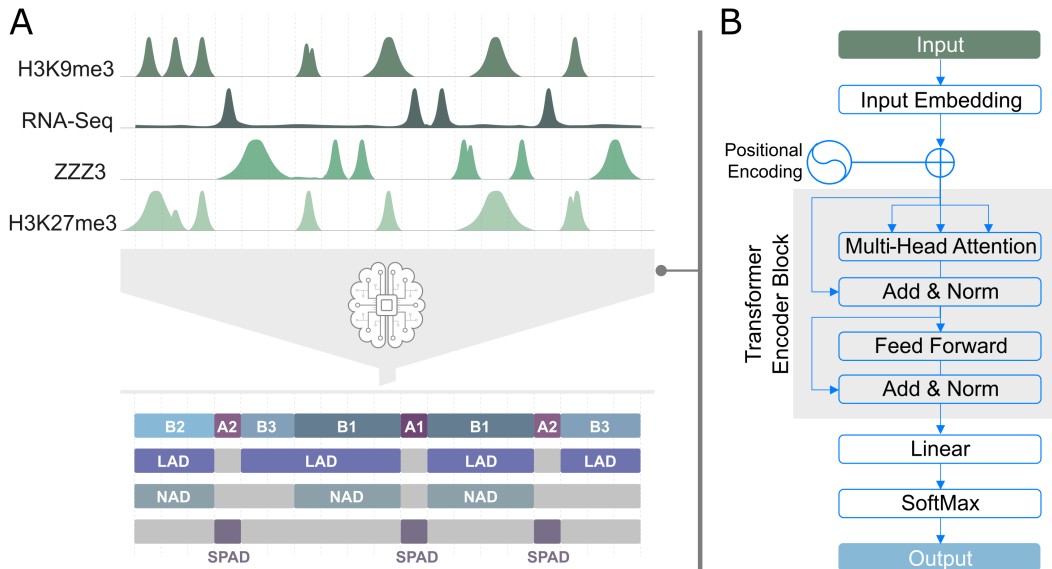

**Fig 1**. **TECSAS workflow for predicting chromatin structure from epigenomic profiles.** (A) Diverse 1D epigenetic tracks (RNA-seq, histone modifications, transcription factor binding) are extracted from the ENCODE portal and segmented into 50kbp loci. The TECSAS deep learning architecture predicts locus-wise structural annotations, including compartments, subcompartments, and potentially other features like LADS, NADS, and SPADS. Prediction is based on learned correlations within the locus's biochemical composition. (B) The TECSAS architecture begins with an input embedding layer, transforming the epigenomic profile into a higher-dimensional representation. A Transformer encoder then analyzes this representation, capturing complex relationships and long-range dependencies within the epigenomic data to understand the structural context. Finally, the output is decoded through a linear layer and a softmax layer, assigning a probability distribution over possible structural annotations for each locus.

profile. Initially, we use the subcompartment annotations from GM12878 derived from Hi-C maps [31] as the target structural information to train TECSAS. The genome is segmented into train, test, and validation sets representing 80%, 10%, and 10% of all the loci, respectively. While the validation set is used to select the model's final set of parameters (parameters resulting in the lowest validation loss during training), all the reported metrics correspond to the model's performance at predicting the test set alone.

The primary output of TECSAS is the prediction of chromatin compartments (A and B) and subcompartments (A1, A2, B1, B2, and B3) for each genomic locus, generating genome-wide structure annotations. However, the model's flexibility allows for predicting additional structural features by modifying the target data and fine-tuning the final layer. For example, by utilizing appropriate training datasets, TECSAS can be adapted to predict the association of loci with specific nuclear bodies, such as Lamina-Associated Domains (LADs), Nucleolus-Associated Domains (NADs), and Speckle-Associated Domains (SPADs). This adaptability makes TECSAS a versatile tool for exploring various aspects of 3D genome organization.

As shown in Fig 1B, TECSAS architecture consists of several key components. First, *an input embedding* which transforms the epigenetic profile into a high-dimensional representation suitable for later interpretation. Second, *a positional fixed signal (positional embedding)* is added to the embedded signal to allow the model to understand the relative positioning between epigenetic signals. The resulting representation of the epigenetic signal is passed through *a transformer encoder* - this layer is in charge of interpreting the epigenetic profile and its internal relationships. Finally, the output from the encoder is decoded by *a linear layer+SoftMax layer* resulting in a probability distribution over the possible structural annotations - while predicting, the model assigns the most likely annotation based on the predicted probabilities.

To evaluate the performance of TECSAS in predicting chromatin subcompartments, we initially trained the model using the well-characterized subcompartment annotations for the GM12878 cell line derived from Hi-C maps. The model utilized

a comprehensive set of epigenomic data from the ENCODE portal, including 11 histone modification ChIP-seq tracks, total and small RNA-seq data, and 140 transcription factor ChIP-seq tracks. For each locus, the input consisted of the signal intensity of these epigenomic features within a 14-locus neighborhood (7 upstream and 7 downstream), capturing the local epigenomic context. The specific hyperparameters used for training TECSAS, such as the number of encoder layers, attention heads, and training epochs, are described in detail in the Methods section. Fig 2A presents the confusion matrix for TECSAS predictions of subcompartments in GM12878, demonstrating high accuracy across all subcompartments. The model achieved an overall accuracy of 0.78, with individual subcompartment accuracies ranging from 0.68 to 0.81. This indicates that each subcompartment possesses a distinct epigenomic signature that TECSAS can effectively learn. Furthermore, the model accurately predicted A/B compartments based on the inferred subcompartments,

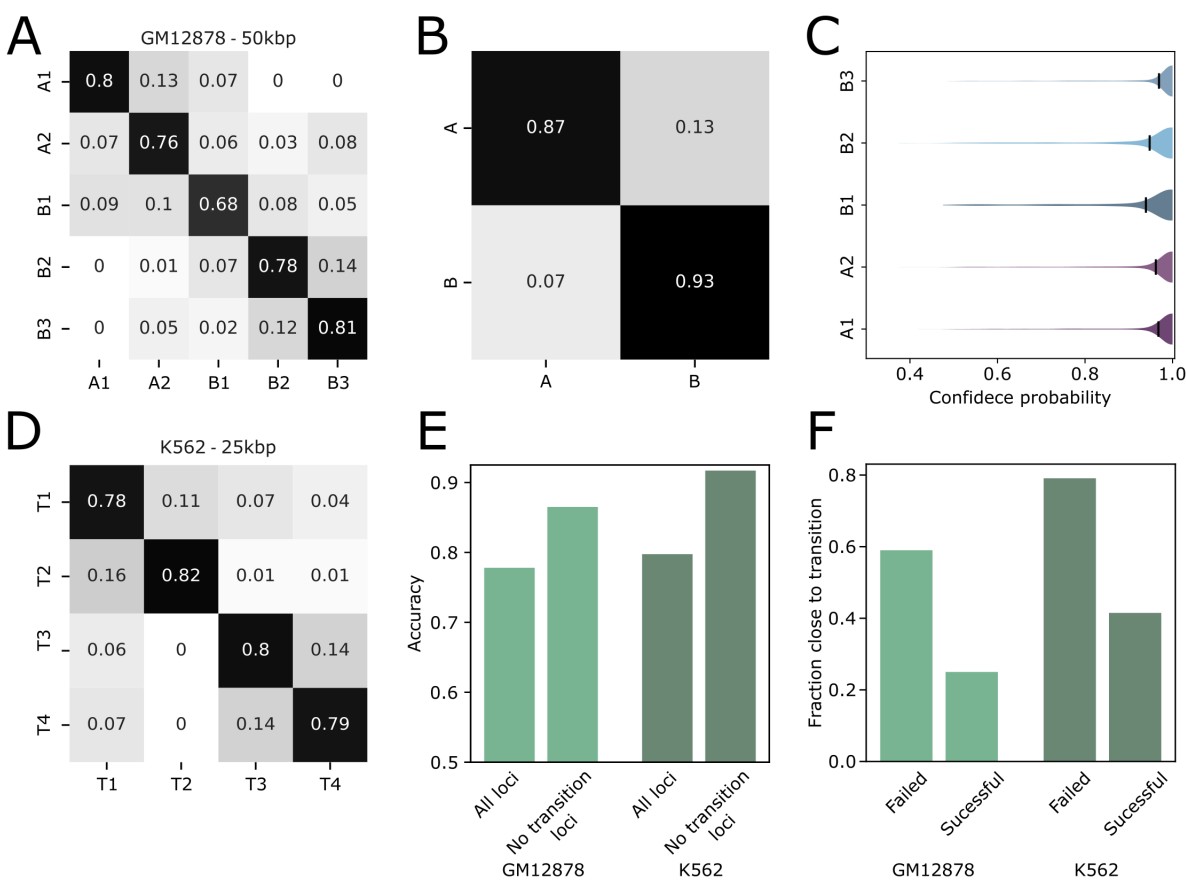

**Fig 2**. **Assessment of TECSAS prediction at 50 kbp and 25kbp resolution for GM12878 and K562 cell lines.** (A) Confusion matrix comparing TECSAS predictions with experimentally derived subcompartment annotations for the GM12878 cell line at 50 kb resolution. The diagonal elements represent the fraction of correctly predicted loci for each subcompartment, highlighting the high accuracy of the model. (B) Confusion matrix for A/B compartment predictions based on the inferred subcompartments in GM12878, demonstrating accurate compartment classification. (C) Distribution of confidence probabilities for each predicted subcompartment in GM12878. B1 and B2 subcompartments exhibit lower average confidence probabilities, reflecting their more complex epigenomic profiles. (D) Confusion matrix comparing TECSAS predictions with subcompartment annotations derived using the SLICE method for the K562 cell line at 25 kb resolution, demonstrating the model's ability to predict subcompartments at higher resolutions. (E) Overall accuracy of TECSAS in predicting subcompartments for GM12878 and K562, comparing performance for all loci and loci excluding transition regions. The exclusion of transition regions significantly improves prediction accuracy for both cell lines. (F) Fraction of successful and failed predictions within transition regions for GM12878 and K562, highlighting the challenges of predicting subcompartments in these regions with mixed epigenomic signatures.

achieving accuracies of 0.87 and 0.93 for A and B compartments, respectively (Fig 2B). This suggests that the epigenomic profiles of A and B compartments are sufficiently distinct to allow for accurate classification. It is important to mention that the overall accuracy between the compartments extracted from different Hi-C methodologies is ≈0.95 (S5 Fig), which means that TECSAS is close to reach the experimental replicate accuracy limit.

TECSAS uses a softmax output layer, which means the output of each node can be related as a probability [32]. Each node represents a different subcompartment. The model predicts the subcompartment for a locus by selecting the node with the highest probability. Using these probabilities, we can extract the confidence of the model at predicting any specific subcompartment, i.e. the higher the prominent probability the more confident the model is at predicting, then, we refer as "confidence probability" as the output probability of the predicted subcompartment. Fig 2C shows that TECSAS has high confidence when predicting the B3 and A1 subcompartments. However, it exhibits lower confidence when predicting the B1 subcompartment. Interestingly, previous research has shown that the B1 subcompartment lacks strong defining characteristics (like specific histone modifications or nuclear body associations [11]). This suggests that B1 has a more complex or less distinct epigenetic profile, making it harder for the model to confidently predict it.

To further assess the model's ability to predict structural annotations at higher resolutions, we trained TECSAS using a set of subcompartments derived from the K562 cell line at 25 kb resolution (S2 Table) using the SLICE method. Despite the increased resolution and a smaller set of input features (124 ChIP-seq experiments), TECSAS achieved a higher overall accuracy of 0.80 in predicting the four K562 subcompartments (Fig 2D). This suggests that the structural annotations derived from SLICE possess identifiable epigenomic profiles, further supporting the link between chromatin's biochemical composition and its 3D organization. This finding also highlights the potential of TECSAS to be applied to higher-resolution data, enabling a more detailed analysis of chromatin structure. Additionally, we observed that TECSAS predictions were less accurate in "transition regions" between subcompartments, defined as regions within four loci of a subcompartment boundary. These regions likely exhibit a mixed epigenomic signature, making it challenging for the model to assign a definitive subcompartment annotation. When excluding these transition regions from the analysis, the prediction accuracy for both GM12878 and K562 subcompartments increased significantly to 0.87 and 0.92, respectively (Fig 2E and 2F). This highlights the importance of considering the gradual nature of epigenomic changes across the genome and the potential for fuzzy structural behavior in transition regions.

## 2.2 Context of the epigenetic profile contributes to the prediction of subcompartments

To investigate the contribution of the epigenomic context to subcompartment prediction accuracy, we compared the performance of TECSAS with PyMEGABASE (PYMB). The PYMB method also only utilizes epigenomic data for predicting structural annotations. Previous benchmarks have shown that PYMB outperforms less complex machine learning techniques such as Random Forest or linear regression [23]. To contextualize TECSAS's performance and align with discussions for rigorous Deep Learning benchmarking strategies in biology [33], we also conducted benchmarks against different machine learning architectures (S11 Fig). In this regard, PYMB is used as a default model for predicting compartments and subcompartments using only 1D tracks (S12 Fig). We used the PYMB benchmark as our baseline to test TECSAS's capability to better understand the nature of epigenetic data using more complex neural network architecture. We first adapted TECSAS to use the same input as PYMB, which consists of discretized signals from histone modification ChIP-seq and RNA-seq experiments, including only two neighboring loci upstream and downstream of the target locus. As shown in Fig 3A, this simplified version of TECSAS achieved an accuracy of 0.62, which is lower than the original TECSAS model (0.78) but still higher than the accuracy of PYMB (0.57). Further, we compare to PYMB the accuracy of TECSAS with different number of experimental tracks, including Histone Modification ChIP-Seq, Transcription Factor ChIP-Seq and RNA-Seq, here we refer as "experiment" to each of these tracks. To do so, we randomly selected 20 sets of 2, ..., 20 experiments and trained the TECSAS, then computing the accuracy on the test set. Fig 3B shows that regardless of the number of experiments used TECSAS outperforms PYMB. This suggests that the Transformer architecture

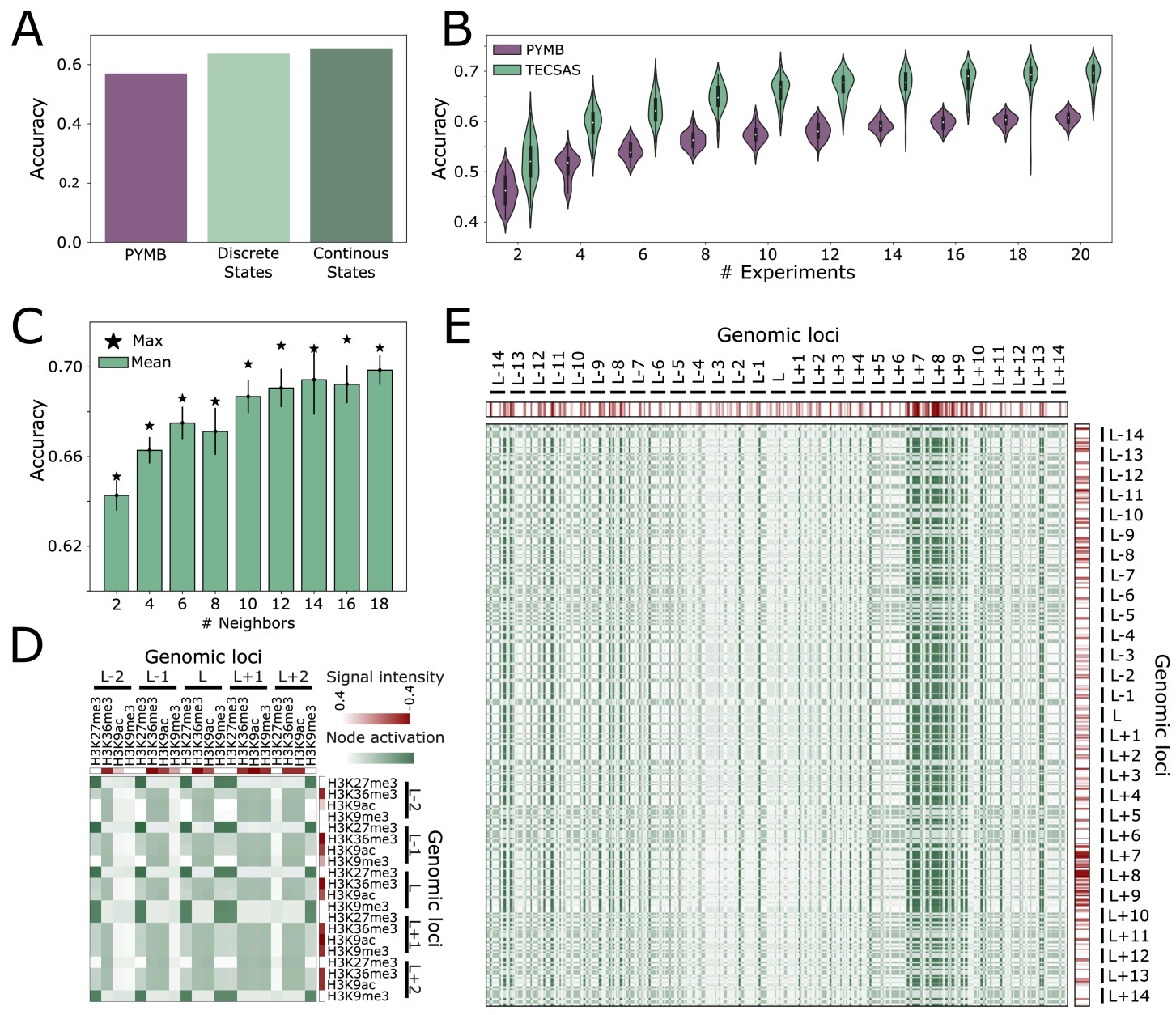

**Fig 3**. **The importance of epigenomic context and long-range interactions for accurate subcompartment prediction with TECSAS.** (A) Comparison of overall accuracy in predicting subcompartments between PyMEGABASE (PYMB) and TECSAS using both discretized and continuous signal intensities for epigenomic features. TECSAS demonstrates higher accuracy even with a limited epigenomic context. (B) Prediction accuracy as a function of the number of input experiments for both PYMB and TECSAS, highlighting the consistent outperformance of TECSAS regardless of the number of features used (p-value $<10^{-9}$ between any PYMB and TECSAS distribution). (C) Mean accuracy of subcompartment predictions with increasing numbers of neighboring loci included in the input, demonstrating the significant improvement in accuracy as the epigenomic context expands. The maximum accuracy achieved is indicated by a star. (D) Subset of the attention map for a locus predicted as A1, showing the activation of nodes (green) corresponding to specific epigenomic features (red) and highlighting the model's focus on relevant patterns within the local epigenomic context. (E) Full attention map for a locus predicted as B1, revealing the importance of long-range interactions and the model's attention to distal regions with enriched epigenetic marks, particularly for marks $\approx$350kbp apart from the locus of interest ($L$).

employed by TECSAS contributes to improved prediction accuracy even when using a limited epigenomic context. Moreover, this benchmark is of particular interest as most of the cell types in the ENCODE portal have available only a small subset of the experiments found for GM12878.

Further analysis revealed that increasing the number of neighboring loci included in the input improves the performance of TECSAS. As shown in Fig 3C (see S10 Fig for PYMB and S14 Fig), the accuracy of the model increases from approximately 0.63 with two neighbors to 0.74 with 18 neighbors, indicating that the structural behavior of a locus is influenced by the epigenomic landscape of a broader genomic region extending up to 900 kb. This observation highlights the importance of capturing long-range dependencies and interactions within the epigenome for accurate prediction of chromatin organization. Notably, the improvement in accuracy with a larger epigenomic context was particularly pronounced for B1 and A1 subcompartments, suggesting that these subcompartments may be more sensitive to the epigenetic state of their surrounding regions. Interestingly, as shown in S1 Fig and S2 Fig, the average epigenetic profile of each subcompartment has a different decay over genomic distance from the locus of interest. The difference between epigenetic profiles between A1 and A2 show how these subcompartments can be identified using the decay of some of the epigenetic marks, similar trend is observed between B1 and B2/3 (S3 Fig and S4 Fig).

To further understand how TECSAS leverages the epigenomic context for subcompartment prediction, we examined the attention maps generated by the model. Unlike PyMEGABASE, which is based on a Potts model and captures fixed pairwise relationships, TECSAS utilizes a Transformer architecture with self-attention. One can consider that the attention layer mechanism can be related to the coupling matrix $J_{ij}$ presented in the Potts model, although a direct comparison is not totally straightforward [32]. The self-attention allows the model to learn complex, conditional relationships where the entire local epigenomic context dynamically modulates the interaction between any two epigenomic features. The importance the model assigns to a specific histone mark is not constant but can change depending on the presence or absence of other marks and transcription factors in the vicinity. While directly visualizing these high-dimensional dependencies is challenging, the attention maps summarize the most influential connections the model uses to inform its predictions.

Fig 3D illustrates a subset of the self-attention map for a locus predicted as A1. The attention map reveals the model's focus on specific patterns within the epigenomic profile. For instance, the enrichment of H3K36me3, a histone modification associated with active transcription, is captured by the activation of nodes corresponding to the H3K36me3 signal. Similarly, the attention map also highlights the depletion of repressive histone marks like H3K9me3 and H3K27me3 downstream of the locus. This demonstrates how TECSAS utilizes the interplay between different epigenetic marks at various genomic distances to inform its predictions.

Further examination of the attention maps reveals the ability of TECSAS to capture long-range interactions within the epigenome. Fig 3E illustrates the full attention map for a locus predicted as B1. While the immediate neighboring loci exhibit relatively low enrichment of epigenetic marks, the attention map highlights the importance of more distal regions, particularly loci $L+7$, $L+8$, and $L+9$, which show higher enrichment of specific histone modifications. This suggests that the structural annotation of a locus can be influenced by the epigenomic landscape of regions located several hundred kilobases apart. The capability to capture these long-range interactions is a key advantage of TECSAS over methods like PYMB, which utilize a more localized epigenomic context and may not fully capture the influence of distal regulatory elements on chromatin organization. By incorporating information from a broader genomic region, TECSAS gains a more comprehensive understanding of the factors that contribute to the 3D structure of chromatin.

Moreover, we expect different 3D behavior of regions where PYMB and TECSAS differ. We explored this possibility by using the OpenMiChroM software [34] to simulate 3D structural ensembles based on the predicted IMR-90 subcompartments from PYMB and TECSAS. One region where the prediction is different for both methods is the chromatin segment chr4:36-37Mbp segment (see S13 Fig for a different segment). As shown in Fig 4A, PYMB predicts it primarily as A-type; in contrast, TECSAS predicts it as B-type, which aligns with the experimental eigenvector. Interestingly, the radial positioning of this segment is significantly different from their respective 3D ensemble of structures (Fig 4B–4C). As

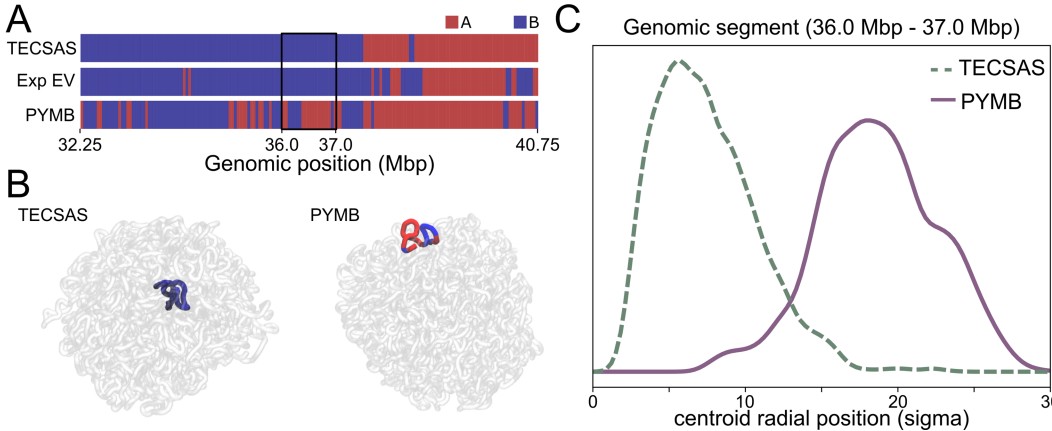

**Fig 4. 3D implications of prediction accuracy on IMR-90.** (A) Compartment annotations from TECSAS, PYMB and experimental Hi-C around the chr4:36-37Mbp segment. (B) Representative structure of chromosome 4 from simulations based on TECSAS and PYMB predictions, highlighting the positioning of the chr4:36-37Mbp segment. (C) Distribution of radial positioning of the chr4:36-37Mbp segment on the simulated ensemble based on TECSAS and PYMB annotations.

expected, the global A and B radial distribution of the chromosome is robust for both sets of predicted annotations (S6 Fig), but local motifs are sensitive to their predicted annotations.

### 2.3 Fine tuning transformer for functional motifs: NADS, LADS, SPADS, Activity profile

Building upon the ability of TECSAS to learn complex relationships between epigenomic features and chromatin structure, we hypothesized that the model could be adapted to predict additional structural information beyond subcompartments. Specifically, we explored the prediction of associations between genomic loci and specific nuclear bodies, such as the nuclear lamina, nucleoli, and speckles. These associations are often characterized as Lamina-Associated Domains (LADs), Nucleolus-Associated Domains (NADs), and Speckle-Associated Domains (SPADs), respectively. To achieve this, we modified the last linear layer of TECSAS to predict whether a given locus belongs to one of these associated domains or its corresponding negative set (non-LAD, non-NAD, non-SPAD). For training, we utilized LAD and NAD annotations for K562, H1, and HCT116 cells derived from DamID experiments, and SPAD annotations for K562 cells derived from TSA-seq experiments, all obtained from the 4DNucleome Data Portal [35]. All associated domain annotations were provided at a 50 kb resolution.

Given that the Transformer encoder, trained on GM12878 data, effectively interprets epigenomic profiles, we focused on training the last linear layer for each type of associated domain, while freezing the transformer encoder parameters. This involved training the linear layer to map the encoded epigenomic information to the specific structural annotation of interest (e.g., LAD or non-LAD). For this analysis, we used only histone modification ChIP-seq data as input, as these assays are widely available across diverse cell types. Fig 5A presents the prediction accuracy for each associated domain, demonstrating high performance with accuracies ranging from 0.78 for non-NADs to over 0.85 for other categories. Notably, the encoder block was trained on GM12878 data, while the last linear layer was trained on combined data from K562, H1, and HCT116 cells. The high accuracy achieved in predicting associated domains suggests that the interpretation of epigenomic features learned by the encoder in GM12878 are transferable to other cell types and for the prediction of other structural annotations. A similar transferability is reported in polymer modeling, where models trained on one cell line, and chromosome can successfully predict experimental Hi-C data in other cell lines.

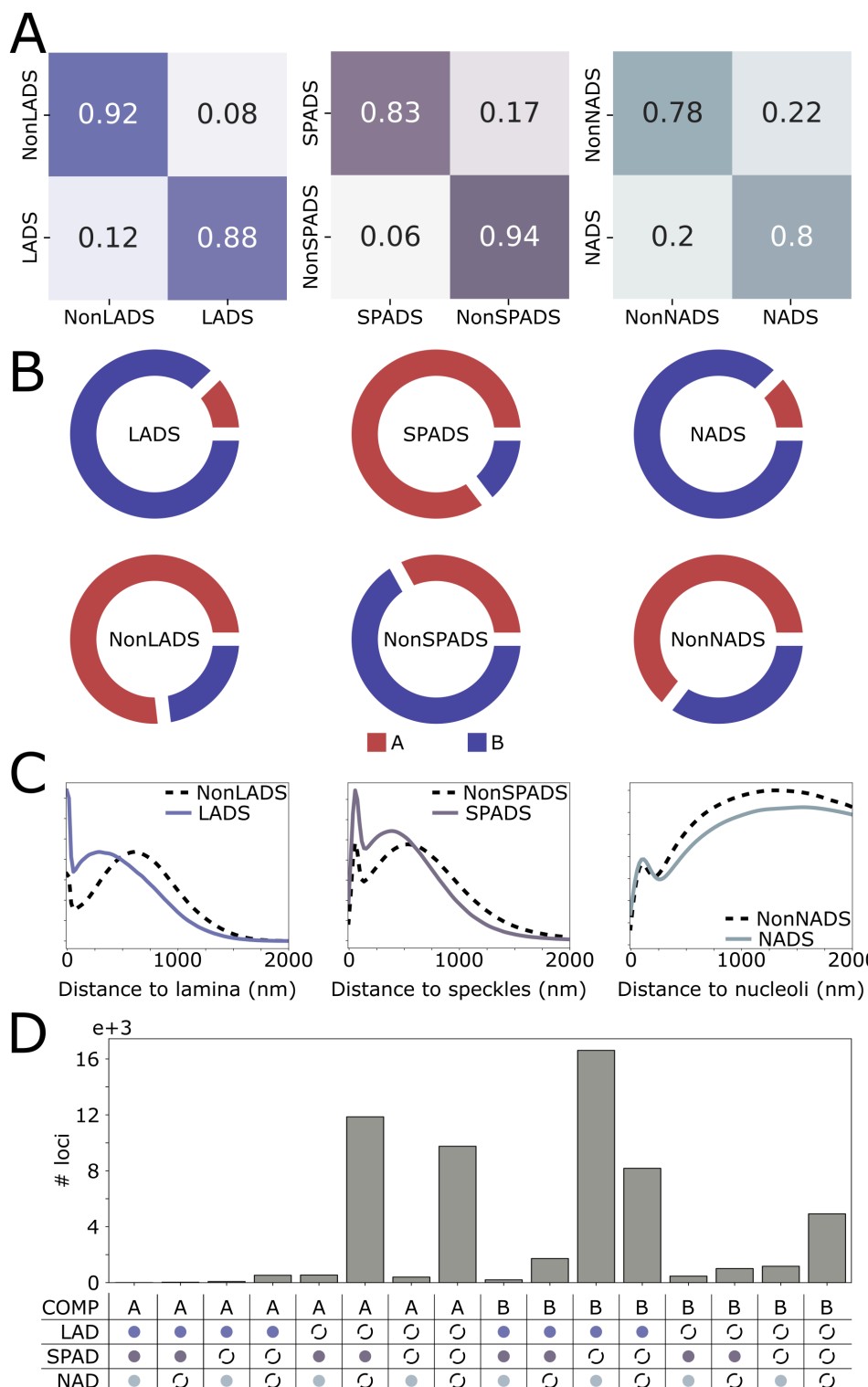

Fig 5. **Prediction of functional structural annotations by TECSAS highlights 3D structural bias due to nuclear body association.** (A) Confusion matrix for predicted LADS, NADS and SPADS against ground truth. (B) Distribution of A and B compartments for IMR-90 for each XAD and nonXAD. (C) Distribution of distance to lamina, speckles and nucleoli for loci predicted as LADS, SPADS and NADS respectively when projected in 3D DNA-tracing experiments [36]. (D) Number of loci in genome predicted as specific combinations of compartment, LAD, SPAD and NAD annotation; solid circles represent XAD and discontinuous circles represents nonXAD.

To further validate the predictions of TECSAS, we applied the model to the IMR-90 cell line, predicting LADs, SPADs, and NADs using histone modification ChIP-seq data as input. We then compared these predictions with A/B compartment annotations derived from the IMR-90 Hi-C data. As expected, LADs and NADs were predominantly found within B compartments, while SPADs were primarily associated with A compartments (Fig 5B). This observation aligns with the reported repressive environment of the nuclear lamina and nucleoli regions and the spatial association of speckles with active transcription. Interestingly, a substantial portion of B compartment regions were predicted to be neither LADs nor NADs. This suggests that these regions are not silenced through association with the lamina or nucleoli, even though they are classified as inactive chromatin. Similarly, a significant fraction of A compartment regions were not predicted as SPADs, indicating that not all active chromatin regions are necessarily localized in spatial proximity with speckles. To further validate TECSAS's generalizability across cell types, we performed a zero-shot experiment where the complete model (encoder and decoder), trained on GM12878's 5-state subcompartment annotations, was applied to predict A/B compartments in the K562 cell line. We used A/B compartment annotations derived from PC1 of Hi-C eigenvector decomposition (ENCODE: ENCFF621AIY) via juicer_tools [37]. TECSAS achieved an accuracy of 0.78 (S9 Fig), compared to 0.87–0.93 in GM12878 (Fig 2B), and outperformed PyMEGABASE (0.68) and simpler baselines (0.60–0.65).

In addition to investigating the spatial distribution of predicted associated domains, we projected the annotations onto 3D genome structures of IMR-90 cells obtained from DNA tracing experiments [36]. These structures provide information about the distance of each chromatin segment to the lamina, speckles, and nucleoli. Fig 5C shows the distribution of distances for each associated domain and its corresponding negative set. As expected, LADs exhibited significantly closer proximity to the lamina compared to non-LADs, and SPADs were located closer to speckles than non-SPADs. NADs showed a slight preference for closer proximity to the nucleoli compared to non-NADs, but the difference was less pronounced than for LADs and SPADs. This suggests that the lamina and speckles may exert a stronger influence on the 3D organization of the genome compared to the nucleoli.

Finally, we analyzed the combined distribution of predicted compartments and associated domains, considering all possible combinations of these annotations (Fig 5D). Five combinations were found to be highly populated (> 4000 loci), with the most frequent A compartment combinations being A-nonLAD-SPAD-nonNAD and A-nonLAD-nonSPAD-nonNAD. This indicates that A compartment regions are primarily differentiated by their association with speckles, while a significant portion does not appear to interact with any of the analyzed nuclear bodies. Similarly, a substantial fraction of B compartment regions were predicted as not associated with the lamina or nucleoli. These findings suggest that nuclear bodies help to organize and shape the 3D structure of chromosomes within the nucleus, contributing to the variety of ways the genome is arranged.

## 3 Discussion and conclusions

This study introduces TECSAS, a deep learning model based on the Transformer architecture, for predicting chromatin structural annotations from one-dimensional epigenomic data. TECSAS utilizes a Transformer encoder to interpret the complex relationships between various epigenetic marks and decode the context of the biochemical composition of the genome. The model achieved high accuracy in predicting subcompartment annotations at both 50 kb and 25 kb resolutions, indicating a strong association between epigenomic profiles and chromatin's structural organization. This finding suggests that changes to the epigenome could be used to directly shape the three-dimensional structure of chromatin.

TECSAS predictions were less accurate in regions transitioning between different subcompartments. These transition regions, characterized by mixed epigenomic signatures, pose a challenge for the model as they do not exhibit a clear association with a single subcompartment. Excluding these transition regions from the analysis significantly improved the prediction accuracy for both GM12878 and K562 subcompartments. This suggests that the epigenomic landscape

undergoes gradual changes across the genome, leading to potentially fuzzy or undefined structural behavior in transition regions. It is worthwhile to mention that even the assumed ground truth experimental data may also include some false positives in the annotations, which may create some noise in the TECSAS predictions.

Compared to PyMEGABASE (PYMB), a previously developed method for predicting structural annotations from epigenomic data, TECSAS demonstrated an improvement in performance (Fig 3). This improvement can be attributed to several factors. First, the Transformer architecture with self-attention mechanisms allows TECSAS to capture complex, non-linear relationships between multiple epigenetic marks, while PYMB relies on a simpler Potts model that primarily captures pairwise interactions. Second, TECSAS incorporates information from a larger neighborhood of loci, enabling the model to account for long-range interactions within the epigenome. Finally, TECSAS utilizes continuous signal intensities for epigenomic features, providing a more nuanced representation of the data compared to the discretized approach used in PYMB.

The versatility of TECSAS extends beyond subcompartment prediction. By fine-tuning the final layer of the model, we successfully predicted the association of genomic loci with specific nuclear bodies, including the lamina, nucleoli, and speckles. The model achieved high accuracy in identifying LADs, NADs, and SPADs, demonstrating its ability to learn transferable relationships between epigenomic features and various structural annotations. The agreement between predicted associated domains and the known functional characteristics of nuclear bodies, such as the association of LADs and NADs with inactive chromatin and SPADs with active transcription, further supports the validity of the model's predictions. Additionally, the analysis revealed heterogeneity within chromatin compartments, with a significant portion of regions not exhibiting a clear association with any of the analyzed nuclear bodies. This suggests that nuclear body association contributes to the diversity of chromatin organization within the nucleus.

In conclusion, this study demonstrates that the biochemical composition of the genome, as reflected in epigenomic data, is highly informative for predicting the three-dimensional organization of chromatin. TECSAS, a deep learning model based on the Transformer architecture, effectively captures the complex relationships between various epigenetic marks and accurately predicts chromatin subcompartments and their association with nuclear bodies. The model's ability to account for long-range interactions and its transferability across cell types highlight its potential as a valuable tool for studying 3D genome organization and its role in gene regulation and other nuclear processes. Future research could explore the application of TECSAS to investigate the functional consequences of nuclear body association and the role of 3D genome organization in various biological contexts, such as different organisms, cell phases, and genetic disorders.

## 4 Methods and materials

### 4.1 Data acquisition and preprocessing

Epigenomic data (histone modification ChIP-seq, transcription factor ChIP-seq, and RNA-seq) were acquired from the ENCODE portal for the GM12878 and K562 cell lines. The initial step utilizes the publicly available pyBigWig software [38] for data fetching.

#### 4.1.1 Data processing.

- **Resolution:** Data were binned into loci of either 50 kbp (GM12878 subcompartment and associated domain prediction) or 25 kbp (K562 subcompartment prediction).
- **Signal Representation:** ChIP-seq signal intensities were expressed as signal p-values. For experiments with multiple replicates, the average signal track was used.
- **Normalization:**
  - **Min-max normalization (chromosome-wise):** The 5th and 95th percentiles were designated as the minimum and maximum values, respectively. This provides a baseline and mitigates outlier influence.
  - **Z-score normalization (chromosome-wise):** Ensures data standardization.

**4.1.2 Input preparation - Preprocess of 1D experimental tracks.** For each target locus, TECSAS input comprised the normalized signal intensities of all epigenomic features within a window of N neighboring loci (both upstream and downstream). The Results section specifies the N value used in each experiment.

## 4.2 TECSAS model architecture and training

TECSAS is a deep learning model utilizing the Transformer architecture. Its key components include:

- **Input Embedding:** The input epigenomic profile is first processed through a linear embedding layer, which transforms the data into a higher-dimensional representation suitable for the Transformer encoder.
- **Positional Encoding:** Positional encoding is added to the embedded input to incorporate information about the relative positions of the loci within the epigenomic profile.
- **Transformer Encoder:** The core of the model is a Transformer encoder with multiple attention heads. The encoder uses self-attention mechanisms to learn complex relationships and dependencies between different epigenomic features across the input loci. This allows the model to capture the context and long-range interactions that influence chromatin structure.
- **Linear and SoftMax Output:** The output of the Transformer encoder is passed through a linear layer, followed by a softmax activation function. The softmax layer outputs a probability distribution over the possible structural annotations, allowing the model to assign the most likely annotation to each locus.

In this paper, we trained the same architecture shown in Fig 1 for several use cases exemplified in Table 1. Independent training, test, and validation sets were generated for models where it was trained and predicted in the same cell type.

**4.2.1 Machine learning implementation.** TECSAS employs a linear layer for the token embedding process, the embedding dimension is set to 128. This is then followed by a transformer encoder block made of two transformer encoder layers with 8 heads each. A linear layer reduces the transformer encoder output to the subcompartment output layers. The dimension of the feedforward layer in the transformer encoder is 64. For training purposes, the dropout rate is set to 1%.These hyperparameters were initially selected empirically, moreover, by using Bayesian Optimization we observe that the selected hyperparameters yield similar accuracy as the best set from the optimization procedure (S1 Table, S7 Fig). PyTorch was used for implementation, and the output layer's activation is processed by a softmax function for activation-to-probability conversion.

For the figures in the main text, the genome is split randomly forming training, validation, testing sets corresponding to 80%, 10% and 10% of all the loci, respectively. In the case of GM12878 (hg19 assembly), this corresponds to 41939 train loci, 5242 test loci and 5242 validation loci, while for all the other cell types (GRCh38 assembly), the data split is distributed as 45984 train loci, 5749 test loci and 5748 validation loci for XADs prediction at 50 kbp and 85652 train loci, 10706 test loci, 10707 validation loci for K562 subcompartments at 25 kbp.

**Table 1. Trained models.** Train, validation and test set where assigned for models trained and tested in the same cell type

| Target | Encoder trained on | Decoder trained on | Predicted on | Experiments | Continuous signal | # of neighbors |
|---|---|---|---|---|---|---|
| Subcompartment | GM12878 | GM12878 | GM12878 | Histone Mod, RNA-Seq | No | 2 |
| Subcompartment | GM12878 | GM12878 | GM12878, IMR-90 | Histone Mod, RNA-Seq | Yes | 2, 14 |
| Subcompartment | GM12878 | GM12878 | GM12878 | TF + HistMod + RNASeq | Yes | 2,4,6,8,10,12,14,16,18 |
| Subcompartment | K562 | K562 | K562 | TF + HistMod + RNASeq | Yes | 14 |
| NADs | GM12878 | K562, H1, HCT116 | K562, H1, HCT116, IMR-90 | TF + HistMod + RNASeq | Yes | 14 |
| LADs | GM12878 | K562, H1, HCT116 | K562, H1, HCT116, IMR-90 | TF + HistMod + RNASeq | Yes | 14 |
| SPADs | GM12878 | K562 | K562, H1, HCT116 | TF + HistMod + RNASeq | Yes | 14 |

The validation set is used to choose the parameters of the model (corresponding to the lowest validation loss during training) and the test set is not used during training. Figures in the SI explore different data splits. All the models in the main text were trained for 70-100 epochs. Source code and tutorials are available on GitHub. The model was trained and tested on AMD Radeon Instinct MI50 32GB GPUs. The Stochastic Gradient Descent optimizer was used for training. The learning rate was initially to 2.5, and it was manually reduced every change checkpoint defined as five epochs. If the training loss is lowered after an epoch, the learning rate change checkpoint is reduced by 1 epoch. TECSAS code is available in https://github.com/ed29rice/TECSAS.git. In the repository training examples and training procedures are included.

### 4.3 Prediction of associated domains

To predict the association of loci with nuclear bodies, we fine-tuned the final layer of TECSAS to output probabilities for LADs, NADs, and SPADs, alongside their corresponding negative sets. We obtained annotations for these domains as follows:

- **LADs and NADs (K562, H1, and HCT116):** Acquired from DamID experiments accessible on the 4D Nucleome Data Portal [35,39]. Relevant experiment IDs include:
  - **K562:** 4DNFIV776O7C
  - **H1:** 4DNFIP6N54B3
  - **HCT116:** 4DNFICCV71TZ
- **SPADs (K562):** Derived from TSA-seq experiments targeting the SON protein (a nuclear speckle marker) [40]. Loci exhibiting signal intensities exceeding the 80th percentile of the genome-wide signal were categorized as SPADs. Experiment ID: 4DNFINI7KVAI.

The fine-tuned model was trained using histone modification ChIP-seq data as input. Performance was assessed on held-out test sets.

## Supporting information

**S1 Fig. GM12878 average epigenetic profile of each of the subcompartments including Histone Modification and RNA-Seq information.**
(TIFF)

**S2 Fig. GM12878 average epigenetic profile of each of the subcompartments including all biochemical markers found in the ENCODE portal.**
(TIFF)

**S3 Fig. Differential average epigenetic profile between A1 and A2, and B1 and B2/3 including all biochemical markers found in the ENCODE portal.**
(TIFF)

**S4 Fig. GM12878 sample epigenetic profile of each of the subcompartments including all biochemical markers found in the ENCODE portal.**
(TIFF)

**S5 Fig. Accuracy between experimental derived compartment annotations from in-situ and dilution Hi-C maps obtained from the ENCODE portal.**
(TIFF)

**S6 Fig. Radial distribution of A and B loci for simulations based on TECSAS and PYMB annotations predictions.**
(TIFF)

**S7 Fig. TECSAS' Hyperparameter Bayesian Optimization progress - odd chromosomes trained on Histone Modifications and RNA-Seq tracks and tested in even chromosomes.**
(TIFF)

**S8 Fig. PYMB and TECSAS accuracy at predicting subcompartment annotations trained in different data splits.**
(TIFF)

**S9 Fig. Confusion matrix comparing TECSAS (trained on GM12878) predictions with compartment annotations derived using the PC1 from Hi-C data for the K562 cell line at 50 kb resolution.**
(TIFF)

**S10 Fig. PYMB performed with different number of neighbors in the input data.**
(TIFF)

**S11 Fig. ML classification methods prediction accuracy of subcompartment on even chromosome on GM12878-hg19, trained on odd chromosomes.**
(TIFF)

**S12 Fig. PYMB and TECSAS (trained on even chromosomes) predictions for subcompartments of different chromosomes in the GM12878 cell line.**
(TIFF)

**S13 Fig. Radial positions of A and B loci for simulations based on TECSAS and PYMB annotations predictions.**
(TIFF)

**S14 Fig. Confusion matrix and accuracies comparing TECSAS (trained on GM12878) predictions on the training set (odd chromosomes) and test set (even chromosomes).**
(TIFF)

**S1 Table. Hyperparameter accuracy comparison using Bayesian Optimization to sample parameter sets.**
(PDF)

**S2 Table. Subcompartment distribution for K562 and GM12878.**
(PDF)

## Acknowledgments

We want to thank Peter Wolynes, Sumitabha Brahmachari, Matheus Mello, and Antonio B. Oliveira Junior for many useful conversations during the development of this work and for all of their comments and suggestions. We also want to thank Tatiana Chavarria Herrera for her help designing and formatting the figures of the article.

## Author contributions

**Conceptualization:** Esteban Dodero-Rojas, Yao Fehlis, Nicholas Malaya, Vinícius G. Contessoto, José N. Onuchic.

**Funding acquisition:** Vinícius G. Contessoto, José N. Onuchic.

**Investigation:** Esteban Dodero-Rojas.

**Methodology:** Esteban Dodero-Rojas, Yao Fehlis, Nicholas Malaya, Vinícius G. Contessoto, José N. Onuchic.

**Software:** Esteban Dodero-Rojas, Angel Mendieta.

**Supervision:** Vinícius G. Contessoto, José N. Onuchic.

**Validation:** Esteban Dodero-Rojas, Angel Mendieta.

**Writing – original draft:** Esteban Dodero-Rojas, Yao Fehlis, Vinícius G. Contessoto, José N. Onuchic.

**Writing – review & editing:** Esteban Dodero-Rojas, Angel Mendieta, Yao Fehlis, Vinícius G. Contessoto, José N. Onuchic.

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
