## [Decision Letter · Decision Letter 0]

8 Jan 2025

PCOMPBIOL-D-24-01194

Epigenetics is all you need: A Transformer to decode chromatin structural compartments from the epigenome

PLOS Computational Biology

Dear Dr. Contessoto,

Thank you for submitting your manuscript to PLOS Computational Biology. After careful consideration, we feel that it has merit but does not fully meet PLOS Computational Biology's publication criteria as it currently stands. Therefore, we invite you to submit a revised version of the manuscript that addresses the points raised during the review process.

Please submit your revised manuscript within 60 days Mar 10 2025 11:59PM. If you will need more time than this to complete your revisions, please reply to this message or contact the journal office at ploscompbiol@plos.org. Please include the following items when submitting your revised manuscript:

We look forward to receiving your revised manuscript.

Kind regards,

Ke Yuan

Guest Editor

PLOS Computational Biology

Shihua Zhang

Section Editor

PLOS Computational Biology

**Journal Requirements:**

At this stage, the following Authors/Authors require contributions: Esteban Dodero-Rojas, Vinícius de Godoi Contessoto, Yao Fehlis, Nicholas Malaya, and Jose Onuchic. Please ensure that the full contributions of each author are acknowledged in the "Add/Edit/Remove Authors" section of our submission form.

5) We have noticed that you have uploaded Supporting Information files, but you have not included a list of legends. Please add a full list of legends for your Supporting Information files after the references list.

Potential Copyright Issues:

i) Figure 1A. Please confirm whether you drew the images / clip-art within the figure panels by hand. If you did not draw the images, please provide (a) a link to the source of the images or icons and their license / terms of use; or (b) written permission from the copyright holder to publish the images or icons under our CC BY 4.0 license. Alternatively, you may replace the images with open source alternatives. See these open source resources you may use to replace images / clip-art:

7) We note that your Data Availability Statement is currently as follows: "All relevant data are within the manuscript and its Supporting Information files.". Please confirm at this time whether or not your submission contains all raw data required to replicate the results of your study. Authors must share the “minimal data set” for their submission. PLOS defines the minimal data set to consist of the data required to replicate all study findings reported in the article, as well as related metadata and methods (https://journals.plos.org/plosone/s/data-availability#loc-minimal-data-set-definition).

8) Please amend your detailed Financial Disclosure statement. This is published with the article. It must therefore be completed in full sentences and contain the exact wording you wish to be published.

2) State what role the funders took in the study. If the funders had no role in your study, please state: "The funders had no role in study design, data collection and analysis, decision to publish, or preparation of the manuscript.".

**Reviewers' comments:**

Reviewer's Responses to Questions

Reviewer #1: The paper “Epigenetics is all you need: A Transformer to decode chromatin structural compartments from the epigenome” from Jose Onuchic’s lab builds upon previous work from the same lab and others, in using a transformer based deep learning architecture, called TECSAS, to infer chromatin structural compartments from multi-omic 1D epigenetic profiles (using here ENCODE as the primary source for input data), which the authors demonstrate represents an improvement in predictive accuracy over PYMB, which is a method from the same lab. The authors are also able to use TECSAS to understand the specific epigenomic features that drive the predictive performance.

The authors address an interesting and important problem, and I agree that exploring it using a transformer based deep learning method makes a lot of sense. The paper is very well written, and the figures are presented to a professional standard. In comparing to PYMB, the authors also elaborate on 3 potential reasons why TECSAS outperforms PYMB. However, I am not entirely satisfied with the amount and type of benchmarking being performed, specially because the authors are benchmarking mainly to their own previous methods. In more detail, my major concerns are as follows:

Major points:

1- Benchmarking to more methods is needed: I realize that the authors have previously published on the same topic, and maybe their previous papers do include more benchmarking, but in the context of this deep-learning paper, I would like to see a lot more benchmarking against much simpler models. I am demanding this following a preprint (see https://www.biorxiv.org/content/10.1101/2024.09.16.613342v3 ) which shows that e.g. fancy deep-learning foundation models do yet not outperform simple linear regression on specific tasks. So, how about if the authors were to use the input data in exactly the same form as used for TECSAS, but instead of feeding it through a transformer architecture, why not use something simpler like Random Forests, a simple MLP or penalized linear regression? Doing so would give the reader a much better understanding of how much ‘deep-learning’ per-se can improve upon ‘non deep-learning’ methods. So, I would like to see more extensive benchmarking against these other ML-methods.

2- Type of benchmarking has limitations: Related to the previous point, I am always uneasy when authors split the set of genomic loci up into training and test sets, when the test sets derive from the same genome. Similar to the preprint above, there have been similar articles published indicating that this form of “validation” can be very biased, i.e. the predictive accuracies are driven by specific features relating to the given cell-type, which do not generalize to other cell-types. Indeed, I notice that in the first analyses in this MS, the authors do not cross-validate a model trained on one cell-type, on another. Rather, it seems that the authors retrain the model for each cell-type separately. Apologies if I misunderstood, but additional clarification to this effect would be really important.

3- Real reason why TECSAS outperforms PYMB: In Discussion, the authors elaborate on 3 reasons why TECSAS outperforms PYMB. However, it is unclear to me that the authors have really shown that all 3 actually are reasons. Maybe it is just 1 or 2 of these? Based on the data presented, it is clear that using larger neighborhoods surrounding loci leads to significant improvements in prediction, and so, because PYMB uses smaller neighborhoods, this is one big reason why TECSAS does better. However, this improvement is therefore unrelated to the ‘deep-learning’ architecture. In this context, it would be important for the authors to clarify WHY PYMB can’t be generalized to use a larger neighborhood surrounding the loci. Could the authors do that and include this ‘enlarged’ PYMB model as an additional method for benchmarking? Assuming the enlarged PYMB model still underperforms in relation to TECSAS, then the improvement is more likely due to the higher order interactions captured by the deep-learning architecture (reason-1). By the same argument, how do we know that the improvement of TECSAS over PYMB is because of the use of continuous signal as input as opposed to binary/ordinal? Could the authors binarize or discretize (3-states) the epigenetic signal and use that as input to TECSAS? Or alternatively, could PYMB be generalized to 3-states? In effect, I think it would be important if the authors could be more clearly pinpoint the actual reason for the improvement, since if it is just the use of a larger neighborhood, then this has nothing to do with the use of a ‘deep-learning’ framework. For a biological-based paper this would not really matter much, but for a methods paper, I really think it is important.

4- Higher order interactions vs pairwise: Related to the previous point, I understand that the Potts Model (PYMB) would only capture pairwise interactions between epigenetic signals right, and that the deep-learning method could capture higher-order interactions that are important for the improvement in prediction. However, the need for higher-order interactions (ie beyond pairwise) for the improved prediction is not clearly borne out from the main figures, in particular Fig.3D-E. Fig.3E only shows the importance of considering large neighborhoods, whilst close inspection of Fig.3D suggests that it is the pairwise interaction of H3K27me3 and H3K9me3 that drives the prediction of this particular locus. In other words, I do apologize if I missed it, but from the main figures I can’t glean a higher-order interaction driving predictive performance, which hence would suggest that maybe the improvement over PYMB has more to do with the use of larger neighborhoods?

5- Fig.3B: some clarification as to which particular experiments were removed and how exactly would help clarify this, specially in the main text where it is not clearly specified. So, for instance, were specific histone marks removed or what exactly is meant by “experiment”?

6- Fig.4: please check labels as I think they are wrong.

7- Section 2.3: In this section, the authors state that “…on GM12878 data, while the last linear layer was trained on combined data from K562, H1, and HCT116 cells. The high accuracy achieved in predicting associated domains suggests that the relationships between epigenomic features and structural annotations learned by TECSAS in GM12878 are transferable to other cell types.” I am not sure this is correct. If you want to show that the structural annotations are transferable to other cell-types, please do NOT use the 2 other cell-types (say H1 and HCT116) in the training process. It would be more convincing if you train on one cell-type only and then demonstrate predictive ability in the other two. See point-2 above.

8- Methods need more clarity: I would like to see more details in the Methods section on how the 10% validation set was used. I presume that the 10% test set is the only truly blinded set, and that the 10% validation set was used to optimize parameters, but which parameters these were is unclear.

9- Some justification for model parameters would be good: In Methods the authors state “TECSAS employs a linear layer for the token embedding process, the embedding dimension is set to 128. This is then followed by a transformer encoder block made of two transformer encoder layers with 8 head each. A linear layer reduces the transformer encoder output to the subcompartment output layers. The dimension of the feedforward layer in the transformer encoder is 64.” Could the authors please explain the rationale for these numbers: 128, two transformer encoder layers with 8 heads each, 64. Can they be varied and if so, how does it affect performance? What features of the input data justify these specific numbers?

Reviewer #2: Summary:

This manuscript introduces TECSAS, a transformer-based deep learning model that predicts chromatin structural annotations from epigenomic data. The model achieves strong performance in predicting subcompartments and nuclear body associations across different cell types. The work is technically sound and well-validated, though there are some areas that could be improved.

Major comments:

The robustness of the model needs clearer validation. The manuscript shows good performance on GM12878 and K562, but should include at least one additional cell type for validation to demonstrate broader applicability.

The authors should better explain how they handle missing data or noise in the epigenomic signals, as this is a common challenge with experimental data. A sensitivity analysis would strengthen the manuscript.

Minor comments:

The comparison with other recent transformer-based methods like MethylGPT and CpGPT should be expanded slightly to position this work in the current landscape.

The visualization in Figure 2C would be more informative if it included error bars to show the variance in confidence probabilities.

The methods section should specify the exact number of training epochs and convergence criteria used.

Some technical terms (e.g., "confidence probability" on page 5) should be defined more precisely.

Reviewer #3: The study by Dodero-Rojas et al. focuses on the development and evaluation of a new transformer-based approach for predicting sub-compartments, leveraging epigenetic context. This method appears to outperform current alternatives and offers intriguing analyses of challenging genomic regions—specifically, those near subcompartmental transitions. Overall, the manuscript is well-structured, and the research question is of substantial interest. However, before further consideration, several key concerns must be addressed. These issues are critical for ensuring the robustness, generalizability, and clarity of your findings. In addition, I have provided some minor points that should be corrected to improve the manuscript’s readability and presentation.

Major Points:

1. Flexibility for Predicting Additional Subcompartments

Recent evidence suggests that the human genome may contain more than the traditionally defined five subcompartments (e.g., Spracklin et al. 2019 https://doi.org/10.1038/s41594-022-00892-7, Zhang et al. 2024 bioRxiv http://dx.doi.org/10.48550/arXiv.2409.14425). It is essential that your method either be directly applicable or easily adaptable to scenarios with a greater number of subcompartments. If the model’s current design inherently limits the number of subcompartments it can predict, this restriction will diminish its appeal to the broader chromatin biology community, where ongoing research increasingly points to a more complex compartmental landscape.

Suggested improvements:

- Please clarify whether the method can be readily extended to predict more than five subcompartments without extensive retraining or modification.

- Consider conducting experiments or analyses where you increase the number of subcompartment classes and report how the model’s performance scales.

- Discuss the methodological adjustments required, if any, and provide guidance on how readers could adapt the approach for future, more fine-grained compartmental annotations.

2. Mitigation of Train-Test Leakage and Proper Evaluation on Unseen Chromosomes

The manuscript states that an 80/10/10 split (training/validation/testing) was used, but it is unclear how the authors avoided overlap between training and test regions, especially given that the model uses contiguous DNA segments as input. Overlapping sequences could lead to train-test leakage and artificially inflated performance. A best practice in genomic modeling tasks is to hold out entire chromosomes for testing, ensuring completely unseen data during evaluation.

Suggested improvements:

- Please describe the procedure in detail, explaining how you guaranteed that no overlap or partial overlap occurs between training and test regions.

- Perform and report an additional experiment where you reserve entire chromosomes (or sufficiently large chromosome segments) as test data.

- Provide metrics demonstrating the model’s performance under these stricter conditions to validate that it generalizes well beyond the training sequences.

3. Clarification of Test Data Usage in Figures and Comparisons

It is not entirely clear which figures and results reflect performance on held-out test sets. The manuscript states that “Performance was assessed on held-out test sets,” but it is ambiguous whether all reported metrics (e.g., in Figures 2 and 3) are derived from test data or a mixture of training, validation, and test sets. Additionally, the number of samples (genomic regions) underlying each figure is not fully disclosed. Finally, if different subsets of the genome were used for comparison to other methods (like PYMB), it is crucial to ensure that comparisons are made on identical, fully withheld test regions to avoid bias and ensure fair comparisons.

Suggested improvements:

- Clearly state which figures and results represent test-only evaluations and indicate the exact data subsets used.

- Report the exact number of genomic regions analyzed for each figure.

- Confirm that comparisons to PYMB and discrete TECSAS were conducted on the same genomic regions held out of training for your method. If this was not the case, please perform comparative analyses on the same withheld set.

- If any reported results were obtained using the entire genome, please repeat those analyses using strictly test-only data and report these new outcomes. This will give readers confidence in the model’s true generalizability.

4. Availability and Quality of Code and Tutorials

The manuscript references a GitHub repository, but a direct link was not found. The availability and quality of associated code, notebooks, and tutorials are essential for transparency, reproducibility, and community adoption of your methodology. Without access to this repository, it is impossible for reviewers and future readers to verify results, adapt the code, or reproduce the analyses.

Suggested improvements:

- Provide a functioning and stable URL to the GitHub repository.

- Ensure that the code is accompanied by clear instructions, documentation, and example scripts that allow other researchers to replicate your results and apply the method to new data.

Minor Concerns:

1. Error Bars and Descriptive Statistics in Figures:

- In Figure 3C, only the mean and maximum values are shown. Please include standard deviations or another measure of variability to offer a more complete statistical description.

- If Figure 3A (and other figures) involves averaging over multiple instances, please also include error bars or confidence intervals to help readers interpret the stability of the reported metrics.

2. In Figure 4, the captions for panels A and C appear to be mixed up. Please correct and ensure that figure panels are described accurately.

3. Typos and Grammatical Errors:

- Page 7, second paragraph from the end: “similar trend is observe between B1 and B2/3” → should be “observed.”

- Page 9, middle of paragraph 2: “ensembles based on the the predicted IMR-90 subcompartments” → remove the second “the.”

- Page 7, beginning: “PyMEGABASE (PYMB), another method that also only utilizes epigenomic data for predicting structural annotations” → Please rephrase this sentence for clarity and grammatical correctness.

A thorough proofreading of the manuscript for English language usage, as well as attention to detail in figure captions and textual consistency, would substantially improve the readability and professionalism of the article.

Reviewer #4: The authors present a paper that uses transformer architecture to predict chromatin sub-compartments annotation from epigenomic data. While the application of transformers is current especially in situations where the surrounding context is vital, the paper can benefit from some clarifications, details, and further experiments that will make it easier to evaluate its claims, suitability, and reach a decision.

Major points

1. Training data and train/test splits: It is not immediately obvious from the text how the 80%, 10%, and 10% splits are generated. If random how do you ensure there is no data leakage between train and test? Is there a case where you can pre-train on one dataset and test on a different dataset to show the model generalises?

2. This point about training and hold-out is valid for all datasets used in the analysis section of the paper. Please provide details how it is done for each results section where you train TECSAS and how you ensure appropriate train/test split?

3. How many examples for each compartment there are? Are some classes unbalanced? This is again unclear from the text/supplementary.

4. Section 2.2: In this section you compare TECSAS to PyMEGABASE. From the text it seems there is another version of the TECSAS model for that task, so to that end you need to make it obvious from the paper how many versions of the model you’ve trained, etc. What are the experiments in Figure B? Maybe provide a table in the supplementary? Again what training data? Train-test splits, etc?

5. Figure 3: Figures A and B can benefit from a statistical test (+ multiple test correction). See comments above about fine-tuning task too: number of data points for each compartment, etc.

6. No actual link to github: Availability is mentioned but no access to the code currently.

Minor points:

1. The explanation of transformers can do with a bit of tightening. The first explanation of transformers perhaps can be rewritten to allow for readers without expert knowledge in the field and the explanation can be extended in Methods.

2. Figures axes labels can be made more readable, use larger fonts, etc. Add axes labels to figure 3E. Please check all your figures to ensure they align with good figure standards.

**Have the authors made all data and (if applicable) computational code underlying the findings in their manuscript fully available?**

Reviewer #1: Yes

Reviewer #2: **No: **

Reviewer #3: **No: **The manuscript references a GitHub repository, but a direct link was not found.

Reviewer #4: **No: **From what I can see the authors wanted to include a link to github but currently the link it's not there. I assume this is just an honest mistake.

PLOS authors have the option to publish the peer review history of their article (what does this mean?). If published, this will include your full peer review and any attached files.

Reviewer #1: No

Reviewer #2: No

Reviewer #3: No

Reviewer #4: No

**Figure resubmission:**
---

## [Decision Letter · Decision Letter 1]

7 May 2025

PCOMPBIOL-D-24-01194R1

Epigenetics is all you need: A Transformer to decode chromatin structural compartments from the epigenome

PLOS Computational Biology

Dear Dr. Contessoto,

Thank you for submitting your manuscript to PLOS Computational Biology. After careful consideration, we feel that it has merit but does not fully meet PLOS Computational Biology's publication criteria as it currently stands. Therefore, we invite you to submit a revised version of the manuscript that addresses the points raised during the review process.

Please submit your revised manuscript within 60 days Jul 07 2025 11:59PM. If you will need more time than this to complete your revisions, please reply to this message or contact the journal office at ploscompbiol@plos.org. Please include the following items when submitting your revised manuscript:

We look forward to receiving your revised manuscript.

Kind regards,

Ke Yuan

Guest Editor

PLOS Computational Biology

Shihua Zhang

Section Editor

PLOS Computational Biology

**Additional Editor Comments :**

Please address major concerns from reviewer 1 and 4.

**Journal Requirements:**

1)  We noted that there are references to supplementary figures in the manuscript; however, there are no corresponding files uploaded to the submission. Please upload them as separate files with the item type 'Supporting Information'. Please add a full list of legends for your Supporting Information files after the references list.

2) Please amend your detailed Financial Disclosure statement. This is published with the article. It must therefore be completed in full sentences and contain the exact wording you wish to be published.

3)Thank you for stating "All the necessary data to reproduce the paper and information is provided in the main text, SI (that includes a GitHub link with the tutorial and data)." Please update your Data Availability Statement in the online submission form to include the GitHub link. Please also ensure that the Supporting Information files are uploaded to the submission.

**Reviewers' comments:**

Reviewer's Responses to Questions

Reviewer #1: I have read the revised version of the paper, but I am afraid to say that I am not satisfied with the responses. In fact, it would appear that the previous work from the authors did NOT perform the requested benchmarks, and hence these benchmarks should be included in this MS following the advice from this preprint (see https://www.biorxiv.org/content/10.1101/2024.09.16.613342v3 ) written by two of the most prominent and experienced bioinformaticians/statisticians. Indeed, as pointed out by Anders & Huber, the DL-application community is not using the correct benchmarks, a concern I deeply share. What is more, the authors claim at the start of section 2.2, that the previous Potts-based models were benchmarked against linear regression and random forests, citing reference [23], but I have inspected ref [23] and this paper does not even contain the words “linear regression” or “random forests”, nor could I find a figure in that paper that provides a direct comparison to these other ML-methods. So, this is very disturbing. I also checked ref [24] and could also not find a comparison to penalized regression or RF. I think it is particularly important, when people are implementing DL methods to benchmark the DL method against a simpler ML-tool such as penalized regression or Random Forests. So, the authors are kindly requested to include these benchmarks in this work to serve as role model for future DL-based studies!

My 2nd major concern was also not addressed. I asked the authors to please provide a form of validation across different cell-types or using separate experiments generated on the same cell-line, but the authors failed to address this. It is simply unacceptable, in light of the above false statements, to claim that this was done in “previous works”. The reader wants to know, if TECSAS is trained on cell-type “A” that it can predict subcompartments in cell-type “B”. Or alternatively, if you train on cell-types “A” and “B” can you predict subcompartments in cell-type “C”. Based on the table provided in the MS, the authors are clearly not performing the validations in the manner that I had advised.

My 3rd major concern was that there was a lack of understanding WHY TECSAS does better than the Potts Model. I argued that based on Fig.3D&E, there is no evidence that the improvement comes from higher-order interaction terms, and indeed Fig.3D does NOT display a higher order interaction. The prediction of the locus in Fig.3D is clearly driven by the pairwise interaction of two marks H3K27me3 and H3K9me3, and whether their interaction is non-linear or linear does not address the issue that this pattern does not involve a 3rd type of histone mark or data-type. Whilst I agree with the authors that the attention-mechanism can in principle capture higher-order interactions, that does not mean that it actually does! Where is the example demonstrating this? After all, the author claim in the MS that their model is interpretable and that they can visualize higher-order interactions, but no figure is shown to demonstrate this.

Related to the previous point, It is also clear that a significant improvement derives mainly from considering larger neighborhoods (Fig.3E), but whether a simpler ML model using larger neighborhoods results in similar prediction accuracies is a key question not explored by the authors. Therefore, the reader has no clear understanding what is driving the improvement. This issue is perfectly addressable, and that the authors are refusing to address it raises concerns, thus compounding earlier concerns about the lack of appropriate benchmarking.

A fifth concern, which I did not mention in my earlier review, is that there is also a lack of a figure that clearly compares methods numerically in terms of some objective evaluation metric. For instance, Fig.4A shows one cherry-picked example where TECSAS performs better than PYMB, but where is the panel comparing validation correlation coefficients with Hi-C data across all cell-lines? You can’t just cherry-pick one region where TECSAS performs better if we could find other regions where PYMB does better.

Reviewer #3: I appreciate the authors’ considerable efforts to refine both the manuscript and its accompanying codebase. The manuscript is now substantially stronger, and my earlier comments have been satisfactorily addressed. I was able to access and run the code without difficulty; however, the online description of the TECSAS instrument remains rather concise. Expanding the tutorials—especially by pinpointing the key steps required for data and intermediary quality control—would greatly aid prospective users in adopting TECSAS with confidence. At present, the codebase still feels somewhat pared‑down, and additional explanatory material would enhance its accessibility.

Reviewer #4: Thank you to the authors for providing an updated version of this manuscript.

Major:

1. Dataset: I understand that the training, test, and validation are split based on chromosomes but if using sequences it’s а good practice to think about the similarity of those sequences to prevent data leakage. Can we quantify this to ensure there isn't any data leakage?

2. Table 1 is very helpful in understanding what you have done but I am still unclear on whether the model generalises. What is the baseline performance of your model? What happens if you use the model trained on GM12878 (both encoder and decoder) and predict K562? That would indicate what generalisable features are learned and to what extent the model can be used when not enough data specific to a cell type is available.

3. Thanks for adding the link to the GitHub but if this is to be widely used there are things needed like guide to installation/requirements file. Tutorial notebooks are lacking comments, so more documentation and details.

Minor:

4. Some of the figures and table references are not working and appear as ? in the text

5. Figure 2C is a bit difficult to read. I get that it’s showing that you are fairly confident with some lower confidence for some compartments but perhaps the visualisation can be improved to make it easier to understand.

**Have the authors made all data and (if applicable) computational code underlying the findings in their manuscript fully available?**

Reviewer #1: Yes

Reviewer #3: Yes

Reviewer #4: Yes

PLOS authors have the option to publish the peer review history of their article (what does this mean?). If published, this will include your full peer review and any attached files.

Reviewer #1: No

Reviewer #3: No

Reviewer #4: No

**Figure resubmission:**
---

## [Decision Letter · Decision Letter 2]

18 Nov 2025

Dear Dr. Contessoto,

We are pleased to inform you that your manuscript 'Epigenetics is all you need: A Transformer to decode chromatin structural compartments from the epigenome' has been provisionally accepted for publication in PLOS Computational Biology.

Best regards,

Shihua Zhang

Section Editor

PLOS Computational Biology

Shihua Zhang

Section Editor

PLOS Computational Biology

Reviewer's Responses to Questions

**Comments to the Authors:**

Reviewer #1: I am broadly speaking satisfied with the revisions. I have no further comments and feel that this work should be published.

**Have the authors made all data and (if applicable) computational code underlying the findings in their manuscript fully available?**

Reviewer #1: Yes

PLOS authors have the option to publish the peer review history of their article (what does this mean?). If published, this will include your full peer review and any attached files.

Reviewer #1: No

---

## [Editor Report · Acceptance letter]

PCOMPBIOL-D-24-01194R2

Epigenetics is all you need: A Transformer to decode chromatin structural compartments from the epigenome

Dear Dr Contessoto,

I am pleased to inform you that your manuscript has been formally accepted for publication in PLOS Computational Biology. Your manuscript is now with our production department and you will be notified of the publication date in due course.

With kind regards,

Judit Kozma
